# Moral Judgments of Human vs. AI Agents in Moral Dilemmas

**DOI:** 10.3390/bs13020181

**Published:** 2023-02-16

**Authors:** Yuyan Zhang, Jiahua Wu, Feng Yu, Liying Xu

**Affiliations:** 1Department of Psychology, School of Philosophy, Wuhan University, Wuhan 430079, China; 2School of Marxism, Tsinghua University, Beijing 100084, China

**Keywords:** artificial intelligence, moral decisions, moral judgment, utilitarianism, deontology

## Abstract

Artificial intelligence has quickly integrated into human society and its moral decision-making has also begun to slowly seep into our lives. The significance of moral judgment research on artificial intelligence behavior is becoming increasingly prominent. The present research aims at examining how people make moral judgments about the behavior of artificial intelligence agents in a trolley dilemma where people are usually driven by controlled cognitive processes, and in a footbridge dilemma where people are usually driven by automatic emotional responses. Through three experiments (*n* = 626), we found that in the trolley dilemma (Experiment 1), the agent type rather than the actual action influenced people’s moral judgments. Specifically, participants rated AI agents’ behavior as more immoral and deserving of more blame than humans’ behavior. Conversely, in the footbridge dilemma (Experiment 2), the actual action rather than the agent type influenced people’s moral judgments. Specifically, participants rated action (a utilitarian act) as less moral and permissible and more morally wrong and blameworthy than inaction (a deontological act). A mixed-design experiment provided a pattern of results consistent with Experiment 1 and Experiment 2 (Experiment 3). This suggests that in different types of moral dilemmas, people adapt different modes of moral judgment to artificial intelligence, this may be explained by that when people make moral judgments in different types of moral dilemmas, they are engaging different processing systems.

## 1. Introduction

Artificial intelligence (AI) technology has developed rapidly in the past few decades and has been widely used in various fields, taking on roles in diagnostic treatment [1], autonomous driving [2], criminal sentencing assessment [3], and wealth management consulting [4]. However, these decisions are closely related to people’s property, as well as physical and mental health, and even involve people’s life and death. Thus, the application of new technology should not only consider its utility but also be cautious about the social impact it may cause. As a result, numerous studies have discussed the problem of the ethics of artificial intelligence’s decisions [5,6,7], focusing on whether the artificial intelligence agents should be responsible for the negative outcomes of decisions they made and how much blame it should bear [8], whether they follow certain biases of their designers, and whether they have intents or motives to commit harmful acts [9,10].

These discussions reflect the fact that people would not perceive AI as mere technological tools used by human agents [11], but sometimes perceived them as moral agents that could act autonomously and be accountable for their behaviors. However, existing research has not reached a consistent conclusion about how people make moral judgments about AI behaviors and decisions. On the one hand, research showed that people tend to make harsher moral judgments of AI agents’ behaviors [12], and resist AI agents making moral decisions [13,14]. On the contrary, there was also evidence that people were more tolerant of AI agents compared to human agents when making moral judgments [6]. In response to these contradictory results, the current study aims to compare people’s moral judgments of human agents and artificial intelligence agents in moral dilemmas and examine what kind of moral norms people apply to AI agents.

### 1.1. Moral Judgments

As a core concept of moral cognition, moral judgment refers to the evaluative judgments that a perceiver makes in response to a moral norm violation, including four major classes of judgment: evaluations, norm judgments, wrongness judgments, and blame judgments, from simple to complex information processing [15]. Evaluations consist of evaluations of good and bad, positive and negative, and represent one of the most basic human responses [16]; evaluative priming can occur without feelings and usually within 1600 milliseconds [17,18]. Norm judgments consist of whether something is permissible, required, forbidden, and so forth based on peoples’ understanding of social rules [15]. Norm judgments are rather different from moral evaluations. Norm judgments invoke the standards against which evaluations are measured and thus set the context for any judgments that are to be called moral [19]. Moral wrongness judgments merge evaluations and norm judgments of intentional actions [15], reflecting an instinctive focus on the negative aspects of events [20,21], such as the tendency of individuals to automatically search for what is wrong with an event [22], and in some cases, people would firmly believe something is wrong even if they could not articulate the reasons for their judgments (moral dumbfounding) [23]. Blame judgments build on all three processes. An initial blame value is hypothesized to be formed from evaluations and wrongness judgments in light of the seriousness of the violated norm [24,25]. Of all moral judgments, blame appears to be the most flexible, complex, and sophisticated, it requires cognitive resources to integrate morally relevant information from multiple sources (e.g., degree of harm, the agent’s causal involvement, intentionality, the agent’s reasons for acting, and counterfactual preventability) to complete the presumption of the blame of the agent [15].

### 1.2. The Dual-Process Theory of Moral Judgment

When faced with moral dilemmas, people may make one of two contradictory choices: utilitarian (or, more broadly, consequentialist) behavior, characterized by the actor making the decision with the aim to maximize benefits and minimize costs across affected individuals [26]; on the contrary, deontological behavior refers to the actor emphasizes responsibilities, rights, and obligations regardless of the outcome [27]. Take the trolley dilemma as an instance (there are five people tied to the track in front of a speeding trolley, you can operate a switch to redirect the trolley onto a side rail, but there is also one person tied to the side rail, would you operate the switch to redirect the trolley?). The act of operating the switch (action) can realize the result of “killing one to save five”, which is a utilitarian act. On the contrary, not operating the switch (inaction) can preserve the life of the “innocent” person on the side rail, that is, never killing to save more lives, which is a deontological act [28]. The two contradictory types of action may elicit completely different moral judgments from the observers.

The dual-process theory of moral judgment explained how the observers’ response process influences their moral judgments about the actors’ utilitarian act or deontological act [29,30]. This theory holds that people’s harsher moral judgments about utilitarian actions are driven by automatic negative emotional responses; while approval of harmful utilitarian actions is driven by controlled cognitive processes [31]. Both automatic emotional responses and more controlled cognitive responses play crucial and, in some cases, mutually competitive roles in individuals’ moral judgments; people’s moral judgments of an act may depend on which process outcompetes in this conflict [29,30]. The most direct and compelling evidence still comes from studies about moral dilemmas. The utilitarian act in the trolley dilemma (i.e., operate the switch to “kill one to save five”) would not evoke the judge’s negative emotional response, at least not very strongly, in this case, the judge makes moral judgments driven by controlled cognitive processes and thus considers the utilitarian choice is more acceptable. In the footbridge dilemma (the actor has to choose between allowing five people to die from a speeding trolley or pushing someone off a footbridge to stop the trolley, saving the five people further down the track, but killing the person pushed), the utilitarian act (i.e., push someone off a footbridge to “kill one to save five”) elicits a prepotent negative emotional response of observers. It could be because the harm, in that case, is more intentional [32,33], more direct [34], and involves intervention and personal force on the victim [32,35,36], or for some other reason. In this case, the negative emotional response that favors deontological choice conflicts with and typically outcompetes the controlled cognitive processes that favor utilitarian choice, so the observer would judge the action of pushing someone off a footbridge as more morally wrong and unacceptable [29,30].

### 1.3. Artificial Intelligence as Moral Agents

With the advancement of artificial intelligence technology and the popularization of its application in daily life, artificial intelligence has gradually been involved in moral events that only humans participated in in the past [11]. In fact, artificial intelligence agents have been regarded as moral agents to a certain extent because of technological developments and people’s perceptions of them. Yagoda and Gillan [37] suggested that technology develops along two dimensions: intelligence and autonomy. Artificial intelligence is capable of autonomously performing various tasks and making decisions without human supervision, and represents the technology of the highest intelligence and autonomy. For example, artificial intelligence can independently complete specific tasks such as recruitment, financial analysis, product recommendation, and medical care [38,39]. Therefore, AI agents are often seen to have some agency [40,41], which refers to the capacities such as communication, planning, and memory [40]. Mind perception is the essence of moral judgment, dimensions of mind perception (agency and experience) map onto moral types (agents and patients), among them, agency, in particular, qualifies entities as moral agents, those who are capable of doing morally good or wrong [42]. In summary, the high intelligence and autonomy of artificial intelligence lead people to perceive AI agents as having mind perception of agency, while agency is linked to moral agents; therefore, AI agents may be regarded as moral agents and thus could be morally judged by people when involved in moral events.

On the other hand, the practical applications of artificial intelligence agents have already sparked a heated debate about their ethics and responsibility as moral agents in social life and academic literature. Take autonomous vehicles (AVs) as a typical example, which are expected to account for 75% of vehicles on the road by 2040 [43]. AVs could increase traffic efficiency and reduce accidents [44,45]; however, not all crashes will be avoided, and some crashes will require AVs to make difficult moral decisions, in cases that involve unavoidable harm—running over pedestrians or sacrificing themselves and its passenger to save them [46]. Car manufacturers and policymakers are currently struggling with these moral dilemmas, in large part because these problems involved a conflict between two moral principles—utilitarianism and deontology. For example, Bonnefon et al. [46] found that people approved of utilitarian AVs (that sacrifice their passengers for the greater good), and would like others to buy them, but they would themselves prefer to ride in AVs that protect their passengers at all costs. If AVs and other autonomous agents do not embed moral principles to guide their decisions in a way that is acceptable to people, there may be some negative outcomes for consumers and manufacturers such as stirring public outrage and discouraging buyers, and thus the world-changing benefits of artificial intelligence agents would also be lost as a result. Accordingly, as we are about to endow countless machines with autonomy, taking AI morality seriously has never been more urgent.

### 1.4. Moral Judgments of Human versus AI Agents

Some studies have explored whether people apply the same moral norms to AI and human agents or not, that is, whether people’s moral judgment of human and AI agents would differ when they make a certain choice (a utilitarian or deontological act). Malle et al. [6] and Voiklis et al. [7] found that people may apply moral norms differentially to humans and AI agents: AI agents are expected—and possibly obligated—to make utilitarian choices. Specifically, participants regarded the act of sacrificing one person to save four (a utilitarian choice) as more permissible for a robot than for a human, a robot that chose this sacrifice was considered morally wrong by far fewer people than a human agent who made that same choice, and human agents were blamed considerably more for taking action than for refraining, whereas robots received almost as much blame for refraining as for taking action. Conversely, Komatsu et al. [47] found that neither the types of agents (human or robot) nor the types of actions (action or inaction) affect the participants’ judgment about moral wrongness for humans and robot agents, that is, people apply the same moral norm to humans and AI agents. Moreover, Bigman and Gray [8] found that people are averse to machines making moral decisions in dilemmas that directly impact human life and death.

### 1.5. The Current Research

Across three studies, we investigated people’s moral judgments of human versus artificial intelligence agents in moral dilemmas. In Experiment 1, participants read about either a human agent or an AI agent who faced a trolley dilemma, and then they read whether the agent’s actual act was action (a utilitarian act) or inaction (a deontological act). In Experiment 2, participants read either a human or an AI agent who faced a footbridge dilemma and their actual actions. In Experiment 3, participants read about both a trolley dilemma and a footbridge dilemma where the agent (a human or an AI agent) performed either a utilitarian act or a deontological act. To measure people’s moral judgments of humans and AI agents, participants rated the morality, permissibility, wrongness, and blameworthiness of the agent’s behavior after reading the scenarios in all three studies (Experiments 1–3).

## 2. Experiment 1

The purpose of Experiment 1 is primarily to examine whether people apply the same or different moral norms and judgments to human agents and AI agents in moral dilemmas that people are typically driven by controlled cognitive processes, using the most classic trolley dilemma as an experimental paradigm. Previous research has found that AI agents, compared with humans, were more commonly expected to make a deontological decision, that is, sacrifice one person for the good of many, and they were blamed more than humans when they refrained from that decision in trolley dilemmas [6,7]. However, it has also been found that people apply the same moral norms and judgments to human and AI agents in trolley dilemmas [47]. Thus, we conducted a validation test about moral judgments about human and AI agents’ behavior in the trolley dilemma in Experiment 1.

### 2.1. Materials and Methods

#### 2.1.1. Participants

One hundred and ninety-five undergraduate students (*M*age = 19.24, *SD* = 1.84, 111 females, 84 males) participated in this study for course credits. This experiment was approved by the Institutional Review Board (IRB) of the authors’ university and all participants gave informed consent.

#### 2.1.2. Materials

Participants read about either a human agent or an AI agent who faced a trolley dilemma [48]. In the trolley problem scenario, the main character had to choose between allowing five people to die from a speeding trolley (inaction) or operating a switch that redirects the trolley onto a side rail, which will save the five people but kill another person (action). A picture describing the scenario was presented below the text at the same time.

#### 2.1.3. Design and Measures

We randomly assigned participants to a 2 (Agent Type: human vs. AI) × 2 (Action: action vs. inaction) between-subjects design. After consenting, participants completed the experiment through Qualtrics, an online survey software program. We experimentally varied the factor Agent Type by describing the main character as either a “railway worker” or an “artificial intelligence program”. We also experimentally varied the factor Action by stating that the agent either did or did not direct the trolley toward the single person. After reading the scenario and learning which action the main character actually chose, participants were asked to indicate the morality, permissibility, wrongness, and blameworthiness of the agent’s behavior (rated on a 100-point slider scale; adopted from [6,7]). Lastly, they answered demographic questions including their age and gender.

### 2.2. Results

Morality. A 2 (Agent Type: human vs. AI) × 2 (Action: action vs. inaction) between-subjects analysis of variance using morality as a dependent measure revealed a significant main effect of agent type, *F* (1, 191) = 4.60, *p* = 0.033, η_p_^2^ = 0.024; specifically, participants rated it less moral in the AI agent condition (*M* = 53.13, *SD* = 2.83, 95% CI [47.56, 58.71]) than those in the human agent condition (*M* = 61.68, *SD* = 2.81, 95% CI [56.14, 67.23]; see Figure 1). However, there was no main effect of action, *F* (1, 191) = 0.13, *p* = 0.724, η_p_^2^ = 0.001, nor an interaction, *F* (1, 191) = 0.003, *p* = 0.960, η_p_^2^ < 0.001.

Permissibility. A 2 (Agent Type: human vs. AI) × 2 (Action: action vs. inaction) between-subjects analysis of variance using permissibility as a dependent measure revealed no main effect of agent type and action, *F* (1, 191) = 0.42, *p* = 0.518, η_p_^2^ = 0.002, and *F* (1, 191) = 0.024, *p* = 0.876, η_p_^2^ = 0.000, nor an interaction, *F* (1, 191) = 0.017, *p* = 0.896, η_p_^2^ = 0.000.

Wrongness. A 2 (Agent Type: human vs. AI) × 2 (Action: action vs. inaction) between-subjects analysis of variance using wrongness as a dependent measure revealed no main effect of agent type and action, *F* (1, 191) = 1.24, *p* = 0.268, η_p_^2^ = 0.006, and *F* (1, 191) = 0.46, *p* = 0.501, η_p_^2^ = 0.002, nor an interaction, *F* (1, 191) = 0.28, *p* = 0.597, η_p_^2^ = 0.001.

Blame. A 2 (Agent Type: human vs. AI) × 2 (Action: action vs. inaction) between-subjects analysis of variance using blame as a dependent measure revealed significant main effects of agent type, *F* (1, 191) = 10.58, *p* = 0.001, η_p_^2^ = 0.052; specifically, participants blamed it more in the AI agent condition (*M* = 43.97, *SD* = 2.69, 95% CI [38.66, 49.28]) than those in the human agent condition (*M* = 31.63, *SD* = 2.67, 95% CI [26.36, 36.91]; see Figure 2). However, there was no main effect of action, *F* (1, 191) = 0.31, *p* = 0.581, η_p_^2^ = 0.002, nor an interaction, *F* (1, 191) = 0.47, *p* = 0.496, η_p_^2^ = 0.002.

All the results of Experiment 1 are summarized in Table 1.

## 3. Experiment 2

In Experiment 1, we found that in the trolley dilemma, people seem to be less concerned about whether the main character in the dilemma makes a utilitarian choice or a deontological choice, but focus more on whether the decision maker is a human or an artificial intelligence agent, people are averse to AI making moral decisions. However, most of the previous studies on moral judgments about AI have been conducted in the context of the trolley dilemmas, but fewer in another classic type of moral dilemma, the footbridge dilemma. Thus, in Experiment 2, we examined how people make moral judgments about human and AI agents’ behavior in the footbridge dilemma.

### 3.1. Materials and Methods

#### 3.1.1. Participants

One hundred and ninety-four undergraduate students (*M*age = 19.18, *SD* = 2.57, 115 females, 79 males) participated in this study for course credits. This experiment was approved by the Institutional Review Board (IRB) of the authors’ university and all participants gave informed consent.

#### 3.1.2. Materials

Participants read about either a human agent or an AI agent who faced a footbridge dilemma [49]. In the footbridge problem scenario, the main character had to choose between allowing five people to die from a speeding trolley (inaction) or pushing someone off a footbridge and onto the path of that speeding trolley, saving the five people further down the track, but killing the person pushed (action). A picture describing the scenario was presented below the text at the same time.

#### 3.1.3. Design and Measures

The procedure of Experiment 2 was very similar to Experiment 1, with only one difference: the moral dilemma was a footbridge problem above instead of a trolley problem. The manipulation of Agent type was identical to that in Experiment 1. We varied the Action by stating that the agent either did or did not push someone off the footbridge onto the path of that speeding trolley. After reading the scenario and learning which action the main character actually chose, participants responded to the same measures of moral judgments and demographic questions as in Experiment 1.

### 3.2. Results

Morality. A 2 (Agent Type: human vs. AI) × 2 (Action: action vs. inaction) between-subjects analysis of variance using morality as a dependent measure revealed a significant main effect of action, *F* (1, 190) = 38.13, *p* < 0.001, η_p_^2^ = 0.167; specifically, participants rated it less moral in the action condition (*M* = 37.52, *SD* = 3.04, 95% CI [31.53, 43.51]) than those in the inaction condition (*M* = 63.77, *SD* = 2.98, 95% CI [57.91, 69.64]). However, there was no main effect of agent type, *F* (1, 190) = 0.20, *p* = 0.654, η_p_^2^ = 0.001, nor an interaction, *F* (1, 190) = 0.08, *p* = 0.775, η_p_^2^ = 0.000.

Permissibility. A 2 (Agent Type: human vs. AI) × 2 (Action: action vs. inaction) between-subjects analysis of variance using permissibility as a dependent measure revealed a significant main effect of action, *F* (1, 190) = 41.89, *p* < 0.001, η_p_^2^ = 0.181; specifically, participants rated it less permissible in the action condition (*M* = 37.66, *SD* = 2.83, 95% CI [32.07, 43.25]) than those in inaction condition (*M* = 63.33, *SD* = 2.78, 95% CI [57.86, 68.81]). However, there was no main effect of agent type, *F* (1, 190) = 0.042, *p* = 0.838, η_p_^2^ = 0.000, nor an interaction, *F* (1, 190) = 3.37, *p* = 0.068, η_p_^2^ = 0.017.

Wrongness. A 2 (Agent Type: human vs. AI) × 2 (Action: action vs. inaction) between-subjects analysis of variance using wrongness as a dependent measure revealed a significant main effect of action, *F* (1, 190) = 40.14, *p* < 0.001, η_p_^2^ = 0.174; specifically, participants rated it more wrong in the action condition (*M* = 64.66, *SD* = 2.85, 95% CI [59.03, 70.29]) than those in inaction condition (*M* = 39.35, *SD* = 2.80, 95% CI [33.83, 44.86]). However, there was no main effect of agent type, *F* (1, 190) = 2.00, *p* = 0.159, η_p_^2^ = 0.010, nor an interaction, *F* (1, 190) = 0.096, *p* = 0.757, η_p_^2^ = 0.001.

Blame. A 2 (Agent Type: human vs. AI) × 2 (Action: action vs. inaction) between-subjects analysis of variance using blame as a dependent measure revealed a significant main effect of action, *F* (1, 190) = 13.95, *p* < 0.001, η_p_^2^ = 0.068; specifically, participants blamed it more in the action condition (*M* = 53.16, *SD* = 2.89, 95% CI [47.46, 58.85]) than those in inaction condition (*M* = 38.05, *SD* = 2.83, 95% CI [32.47, 43.63]). However, there was no main effect of agent type, *F* (1, 190) = 0.007, *p* = 0.935, η_p_^2^ = 0.000, nor an interaction, *F* (1, 190) = 0.24, *p* = 0.626, η_p_^2^ = 0.001.

All the results of Experiment 2 are summarized in Table 2.

## 4. Experiment 3

In Experiments 1 and 2, we found that in the trolley dilemma people were more concerned with who made the decision and less concerned with whether it was a utilitarian or deontological decision; more specifically, people were averse to AI making moral decisions about human life and death. In contrast, in the footbridge dilemma, people pay less attention to whether the decision maker is a human or an artificial intelligence agent but pay more attention to whether the agent makes a utilitarian decision or a deontological decision; more specifically, people were averse to a utilitarian decision in the footbridge dilemma regardless of whether the decision maker is a human or an AI agent. To verify the consistency of the results of Experiments 1 and 2, and to further explore the differences in people’s moral judgments about human and AI agents’ behavior in different types of dilemmas, we conducted a within-subject experiment in Experiment 3, using Dilemma Type as a within-subject factor.

### 4.1. Materials and Methods

#### 4.1.1. Participants

Two hundred and thirty-six undergraduate students (*M*age = 18.93, *SD* = 1.67, 147 females, 89 males) participated in this study for course credits. This experiment was approved by the Institutional Review Board (IRB) of the authors’ university and all participants gave informed consent.

#### 4.1.2. Materials

The moral dilemma scenarios were identical to those in Experiments 1 and 2.

#### 4.1.3. Design and Measures

The experiment utilized a 2 (Dilemma Type: trolley dilemma vs. footbridge dilemma) × 2 (Agent Type: human vs. AI) × 2 (Action: action vs. inaction) mixed design with Dilemma Type as a within-subject factor and both Agent Type and Action as between-subject factors. We randomly assigned participants to one of each condition. After consenting, participants completed the experiment through Qualtrics, an online survey software program. Participants read both two scenarios, with half the sample reading the trolley dilemma first and another half the sample reading the footbridge dilemma first. The manipulations of Agent Type and Action were identical to those in Experiments 1 and 2. In the end, participants responded to the same measures of moral judgments and demographic questions as in Experiments 1 and 2.

### 4.2. Results

Morality. A 2 (Dilemma Type: trolley dilemma vs. footbridge dilemma) × 2 (Agent Type: human vs. AI) × 2 (Action: action vs. inaction) analysis of variance, using morality as a dependent measure, dilemma type as a within-subjects factor, agent type and action as between-subjects factors, revealed a significant main effect of dilemma type, *F* (1, 232) = 8.35, *p* = 0.004, η_p_^2^ = 0.035; specifically, participants rated it less moral in the footbridge dilemma condition (*M* = 46.80, *SD* = 1.84, 95% CI [43.17, 50.43]) than those in the trolley dilemma condition (*M* = 52.00, *SD* = 1.74, 95% CI [48.58, 55.42]). Additionally, there was also a significant main effect of action, *F* (1, 232) = 26.68, *p* < 0.001, η_p_^2^ = 0.103; specifically, participants rated it less moral in the action condition (*M* = 41.41, *SD* = 2.18, 95% CI [37.11, 45.70]) than those in the inaction condition (*M* = 57.39, *SD* = 2.20, 95% CI [53.06, 61.72]). However, there was no main effect of agent type, *F* (1, 232) = 0.71, *p* = 0.400, η_p_^2^ = 0.003.

There was a significant dilemma type × action interaction, *F* (1, 232) = 12.60, *p* < 0.001, η_p_^2^ = 0.052; specifically, in the action condition, participants rated it less moral in the footbridge dilemma condition (*M* = 35.62, *SD* = 2.59, 95% CI [30.51, 40.73]) than in the trolley dilemma condition (*M* = 47.20, *SD* = 2.45, 95% CI [42.38, 52.02]). In the inaction condition, morality did not vary with the dilemma type condition, *F* (1, 232) = 0.216, *p* = 0.643, η_p_^2^ = 0.001. None of the other interaction effects was significant (smallest *p* = 0.156).

Permissibility. A 2 (Dilemma Type: trolley dilemma vs. footbridge dilemma) × 2 (Agent Type: human vs. AI) × 2 (Action: action vs. inaction) analysis of variance, using permissibility as a dependent measure, dilemma type as a within-subjects factor, agent type and action as between-subjects factors, revealed a significant main effect of dilemma type, *F* (1, 232) = 11.37, *p* = 0.001, η_p_^2^ = 0.047; specifically, participants rated it less permissible in the footbridge dilemma condition (*M* = 48.08, *SD* = 1.76, 95% CI [44.61, 51.56]) than those in the trolley dilemma condition (*M* = 54.33, *SD* = 1.65, 95% CI [51.07, 57.59]). Additionally, there was also a significant main effect of action, *F* (1, 232) = 12.28, *p* = 0.001, η_p_^2^ = 0.050; specifically, participants rated it less permissible in the action condition (*M* = 46.17, *SD* = 2.02, 95% CI [42.19, 50.16]) than those in the inaction condition (*M* = 56.24, *SD* = 2.04, 95% CI [52.22, 60.26]). The main effect of agent type was marginally significant, *F* (1, 232) = 3.58, *p* = 0.060, η_p_^2^ = 0.015; specifically, participants rated it less permissible in the AI agent condition (*M* = 48.49, *SD* = 2.05, 95% CI [44.45, 52.52]) than those in the human agent condition (*M* = 53.93, *SD* = 2.02, 95% CI [49.96, 57.90]).

There was a significant dilemma type × action interaction, *F* (1, 232) = 14.35, *p* < 0.001, η_p_^2^ = 0.058; specifically, in the footbridge dilemma condition, participants rated it less permissible in the action condition (*M* = 39.54, *SD* = 2.48, 95% CI [34.65, 44.43]) than those in the inaction condition (*M* = 56.63, *SD* = 2.50, 95% CI [51.70, 61.56]). In the trolley dilemma condition, permissibility did not vary with the action condition, *F* (1, 232) = 0.85, *p* = 0.358, η_p_^2^ = 0.004. The interaction effect of dilemma type × agent type was marginally significant, *F* (1, 232) = 2.93, *p* = 0.088, η_p_^2^ = 0.012; specifically, in the trolley dilemma, participants rated it less permissible in the AI agent condition (*M* = 50.03, *SD* = 2.36, 95% CI [45.38, 54.68]) than in the human agent condition (*M* = 58.64, *SD* = 2.32, 95% CI [54.06, 63.21]), *F* (1, 232) = 6.77, *p* = 0.010, η_p_^2^ = 0.028. In the footbridge condition, permissible did not vary with agent type condition, *F* (1, 232) = 0.414, *p* = 0.521, η_p_^2^ = 0.002 (see Figure 3). There was also a marginal significant agent type × action interaction, *F* (1, 232) = 3.21, *p* = 0.075, η_p_^2^ = 0.014; specifically, in the human agent condition, participants rated it less permissible in the action condition (*M* = 46.32, *SD* = 2.83, 95% CI [40.75, 51.89]) than in the inaction condition (*M* = 61.53, *SD* = 2.87, 95% CI [55.87, 67.20]), *F* (1, 232) = 14.25, *p* < 0.001, η_p_^2^ = 0.058. In the AI agent condition, permissible did not vary with action, *F* (1, 232) = 1.44, *p* = 0.231, η_p_^2^ = 0.006. There was no dilemma type × action interaction, *F* (1, 232) = 0.005, *p* = 0.942, η_p_^2^ = 0.000.

Wrongness. A 2 (Dilemma Type: trolley dilemma vs. footbridge dilemma) × 2 (Agent Type: human vs. AI) × 2 (Action: action vs. inaction) analysis of variance, using wrongness as a dependent measure, dilemma type as a within-subjects factor, agent type and action as between-subjects factors, revealed a significant main effect of dilemma type, *F* (1, 232) = 11.11, *p* = 0.001, η_p_^2^ = 0.046; specifically, participants rated it more wrong in the footbridge dilemma condition (*M* = 54.14, *SD* = 1.80, 95% CI [50.60, 57.67]) than those in the trolley dilemma condition (*M* = 47.54, *SD* = 1.76, 95% CI [44.07, 51.01]). Additionally, there was also a significant main effect of action, *F* (1, 232) = 16.69, *p* < 0.001, η_p_^2^ = 0.067; specifically, participants rated it more wrong in the action condition (*M* = 56.87, *SD* = 2.08, 95% CI [52.78, 60.97]) than those in the inaction condition (*M* = 44.81, *SD* = 2.10, 95% CI [40.67, 48.94]). However, there was no main effect of agent type, *F* (1, 232) = 0.276, *p* = 0.600, η_p_^2^ = 0.001.

There was a significant dilemma type × action interaction, *F* (1, 232) = 5.90, *p* = 0.016, η_p_^2^ = 0.025; specifically, in the action condition, participants rated it more wrong in the footbridge dilemma condition (*M* = 62.58, *SD* = 2.53, 95% CI [57.60, 67.56]) than those in the trolley dilemma condition (*M* = 51.17, *SD* = 2.48, 95% CI [46.28, 56.06]), *F* (1, 232) = 16.74, *p* < 0.001, η_p_^2^ = 0.067. In the inaction condition, wrongness did not vary with the dilemma type, *F* (1, 232) = 0.41, *p* = 0.525, η_p_^2^ = 0.002. None of the other interaction effects was significant (smallest *p* = 0.167).

Blame. A 2 (Dilemma Type: trolley dilemma vs. footbridge dilemma) × 2 (Agent Type: human vs. AI) × 2 (Action: action vs. inaction) analysis of variance, using blame as a dependent measure, dilemma type as a within-subjects factor, agent type and action as between-subjects factors, revealed a significant main effect of dilemma type, *F* (1, 232) = 13.67, *p* < 0.001, η_p_^2^ = 0.056; specifically, participants blame it more in the footbridge dilemma condition (*M* = 49.73, *SD* = 1.77, 95% CI [46.25, 53.21]) than those in the trolley dilemma condition (*M* = 42.63, *SD* = 1.79, 95% CI [39.09, 46.16]). Additionally, there was also a significant main effect of action, *F* (1, 232) = 19.12, *p* < 0.001, η_p_^2^ = 0.076; specifically, participants blamed it more in the action condition (*M* = 52.72, *SD* = 2.11, 95% CI [48.57, 56.88]) than those in the inaction condition (*M* = 39.63, *SD* = 2.13, 95% CI [35.44, 43.82]). There was no main effect of action type, *F* (1, 232) = 0.001, *p* = 0.982, η_p_^2^ = 0.000.

There was a significant dilemma type × action interaction, *F* (1, 232) = 10.68, *p* = 0.001, η_p_^2^ = 0.044; specifically, in the footbridge dilemma condition, participants blamed it more in the action condition (*M* = 59.42, *SD* = 2.49, 95% CI [54.52, 64.31]) than those in the inaction condition (*M* = 40.04, *SD* = 2.51, 95% CI [35.10, 44.98]), *F* (1, 232) = 30.11, *p* < 0.001, η_p_^2^ = 0.115. In the trolley dilemma, blame did not vary with action, *F* (1, 232) = 3.62, *p* = 0.058, η_p_^2^ = 0.015. The interaction effect of dilemma type × agent type was marginally significant, *F* (1, 232) = 3.54, *p* = 0.061, η_p_^2^ = 0.015. None of the other interaction effects was significant (smallest *p* = 0.162).

All the results of Experiment 3 are summarized in Table 3.

## 5. Discussion

In the trolley dilemma (Experiment 1), the agent type rather than the actual action influenced people’s moral judgments. Specifically, participants rated AI agents’ behavior as more immoral and deserving of more blame than humans’ behavior, regardless of whether they act utilitarianly or deontologically. Conversely, in the footbridge dilemma (Experiment 2), the actual action rather than the agent type influenced people’s moral judgments. Specifically, participants rated action (a utilitarian act) as less moral and permissible and more morally wrong and blameworthy than inaction (a deontological act), regardless of whether the actor is a human or an AI agent. The result of Experiment 3 provided a converging pattern that, in the trolley dilemma, agent type influenced people’s moral judgments: participants rated human agents’ behavior as more permissible than AI agents’ behavior. On the contrary, in the footbridge dilemma, only action influenced people’s moral judgments, participants rated action as less moral and permissible and more morally wrong and blameworthy than inaction; agent type did not influence people’s moral judgments in this dilemma. There was one small difference between Experiment 3 and Experiment 1. In Experiment 1, only agent type influenced people’s moral judgments, while in Experiment 3, people were interested in both the difference between humans and AI and the action versus inaction; people rated the action as less moral and more morally wrong than inaction in the trolley dilemma. It may be explained that in Experiment 3, people read both the two types of moral dilemmas and people’s focus may be influenced by the previous scenario when making moral judgments about the second scenario.

Overall, these findings revealed that, in the trolley dilemma, people are more interested in the difference between humans and AI agents than action versus inaction. Conversely, in the footbridge dilemma, people are more interested in action versus inaction. It may be explained that people made moral judgments driven by different response processes in these two dilemmas—controlled cognitive processes occur often in response to dilemmas such as the trolley dilemma and automatic emotional responses occur often in response to dilemmas such as the footbridge dilemma [30]. Thus, in the trolley dilemma, controlled cognitive processes may drive people’s attention to the agent type and make the judgment that it is inappropriate for AI agents to make moral decisions. In the footbridge dilemma, the action of pushing someone off a footbridge may evoke a stronger negative emotion than the action of operating a switch in the trolley dilemma. Driven by these automatic negative emotional responses, people would focus more on whether the agents did this harmful act, and judged this harmful act less acceptable and more morally wrong.

However, it should be noted that our work presents some limitations and offers several avenues for future research. Firstly, the current study only examined how people make moral judgments about humans and AI agents, but did not investigate the underlying psychological mechanism. Thus, all interpretations of the results are speculations. Future research could further explore the reason why people are reluctant to AI agents making moral decisions in the trolley dilemma, why people apply the same moral norms to humans and AI agents in the footbridge dilemma, and why people show different patterns of moral judgment in the trolley dilemma and the footbridge dilemma. Previous research has provided us with some pointers. For example, interpretability and consistency of behaviors would increase people’s acceptance of AI [50,51]; increased anthropomorphism of an autonomous agent would mitigate blame for an agent’s involvement in an undesirable outcome [52]. Individual differences including personality [53,54], development experiences [55,56,57], and cultural background [58] may also influence people’s attitudes toward AI agents. Second, to exclude the potential influence of individual differences between Experiments 1 and 2, we conducted Experiment 3 with a within-subjects design, participants were asked to read both the two scenarios; however, the processing system activated by the first scenario may influence the participants’ judgment about the subsequent scenario. For example, participants who read the footbridge dilemma first may be interested in whether the character acted or not due to the strong negative emotion, this emotion may drive people to focus on the character’s action in the subsequent trolley dilemma, just like they did in the footbridge dilemma. Future research could consider other method approaches to exclude the effects of individual differences and order effects.

Last but not least, we examined people’s moral judgments of humans and AI agents using the most classical and traditional dilemma paradigms. However, recent studies have discussed the method limitations of these traditional dilemma paradigms. For example, Gawronski et al. [59] claimed that the conceptual meaning of responses in the traditional paradigm is ambiguous because the central aspects of utilitarianism and deontology—consequences and norms—are not manipulated. This shortcoming may undermine empirical findings such as the problem that the traditional moral paradigms could not measure the general action/inaction biases of participants. Given this, there might be other explanations for the current results, such as the inclination to be overall acceptable to any behavioral proposals. Specifically, there might be a stronger overall inclination to accept the action choices in the trolley dilemma than in the footbridge dilemma. This is an alternative explanation that should be tested in future studies. To resolve these limitations of traditional dilemma paradigms, Gawronski et al. presented a multinomial model (the CNI model) that allows researchers to quantify sensitivity to consequences, sensitivity to moral norms, and a general preference for inaction versus action irrespective of consequences and norms in responses to moral dilemmas (for details, see [59,60]). More recently, however, the limitations of the CNI model were also been discussed (for details, see [61,62]). As a result, Liu and Liao [63,64] developed a new algorithm—the CAN—to address the methodological limitations of the CNI model and fix these limitations (for details, see [63,64,65]). We suggest that further research could examine people’s moral judgments of humans and AI agents using these newly developed approaches and make a comparison with the current results that come from the traditional dilemma paradigms.

## Figures and Tables

**Figure 1 behavsci-13-00181-f001:**
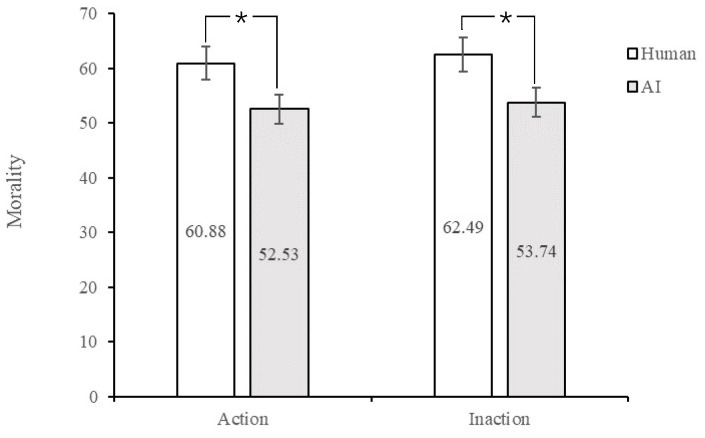
Rates of morality on human and AI agents. Note. * *p* < 0.05.

**Figure 2 behavsci-13-00181-f002:**
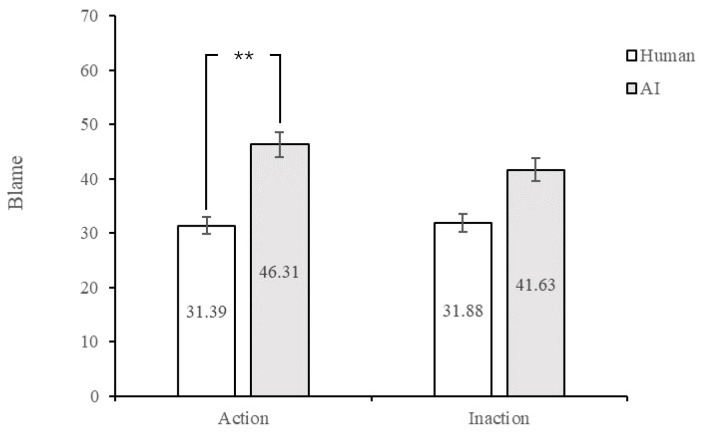
Rates of blame on human and AI agents. Note. ** *p* < 0.01.

**Figure 3 behavsci-13-00181-f003:**
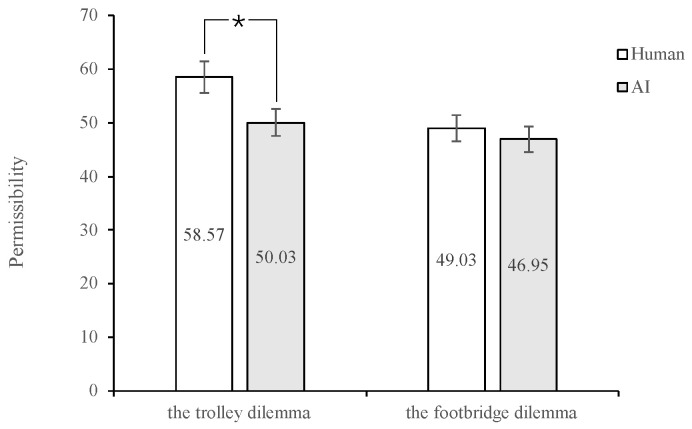
Rates of permissibility on human and AI agents. Note. * *p* < 0.05.

**Table 1 behavsci-13-00181-t001:** Rates of morality, permissibility, wrongness, and blame on human and AI agents in Experiment 1.

Agent Type	Action	Morality	Permissibility	Wrongness	Blame
human agents	action	60.88 ± 3.97	61.14 ± 3.97	44.63 ± 4.06	31.39 ± 3.78
	inaction	62.49 ± 3.97	60.00 ± 3.97	44.04 ± 4.06	31.88 ± 3.78
AI agents	action	52.53 ± 3.90	58.04 ± 3.89	51.31 ± 3.98	46.31 ± 3.71
	inaction	53.74 ± 4.10	57.93 ± 4.10	46.41 ± 4.19	41.63 ± 3.90
agent type		*			**
action					
agent type * action					

Note. * *p* < 0.05, ** *p* < 0.01.

**Table 2 behavsci-13-00181-t002:** Rates of morality, permissibility, wrongness, and blame on human and AI agents in Experiment 2.

Agent Type	Action	Morality	Permissibility	Wrongness	Blame
human agents	action	35.96 ± 4.27	40.90 ± 3.99	68.10 ± 4.02	52.33 ± 4.06
	inaction	63.43 ± 4.23	59.39 ± 3.95	41.55 ± 3.97	39.20 ± 4.02
AI agents	action	39.09 ± 4.32	34.43 ± 4.03	61.21 ± 4.06	53.98 ± 4.11
	inaction	64.12 ± 4.19	67.38 ± 3.91	37.14 ± 3.93	36.90 ± 3.98
agent type					
action		**	**	**	**
agent type * action					

Note. * *p* < 0.05, ** *p* < 0.01.

**Table 3 behavsci-13-00181-t003:** Rates of morality, permissibility, wrongness, and blame on human and AI agents in Experiment 3.

Dilemma Type	Agent Type	Action	Morality	Permissibility	Wrongness	Blame
the trolley dilemma	human agents	action	48.74 ± 3.42	54.46 ± 3.25	54.03 ± 3.46	47.67 ± 3.53
		inaction	60.42 ± 3.47	62.81 ± 3.31	40.36 ± 3.52	34.03 ± 3.59
	AI agents	action	45.66 ± 3.50	51.16 ± 3.34	48.31 ± 3.55	44.40 ± 3.62
		inaction	53.17 ± 3.50	48.90 ± 3.34	47.47 ± 3.55	44.40 ± 3.62
the footbridge dilemma	human agents	action	33.87 ± 3.62	38.18 ± 3.47	65.36 ± 3.53	61.97 ± 3.47
		inaction	59.78 ± 3.68	60.25 ± 3.53	46.71 ± 3.59	41.17 ± 3.53
	AI agents	action	37.36 ± 3.71	40.90 ± 3.56	59.79 ± 3.62	56.86 ± 3.56
		inaction	56.19 ± 3.71	53.00 ± 3.56	44.69 ± 3.62	38.91 ± 3.56
dilemma type			**	**	**	**
agent type				*p* = 0.060		
action			**	**	**	**
dilemma type * agent type				*p* = 0.088		*p* = 0.061
dilemma type * action			**	**	*	**
agent type * action				*p* = 0.075		
dilemma type * agent type * action						

Note. * *p* < 0.05, ** *p* < 0.01.

## Data Availability

The authors will share data from the study upon reasonable request to the corresponding author.

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
