# Peer review of "Moral Judgments of Human vs. AI Agents in Moral Dilemmas"

_behavsci, 2023, doi:10.3390/bs13020181_

Round 1
Reviewer 1 Report
Dear Authors and Editor,
I analyzed the material sent and made a few observations:
Artificial intelligence is a very relevant topic that will continue to grow in the coming years.
Summary: meets the requirements for a summary.
Introduction: the text contextualizes the research, of course, it was elaborated with four subtopics that function as a theoretical foundation on the subject of the study.
Methods: each of the experiments is presented in a very clear and detailed way, separately.
Results: are also presented separately by experiment; as it is a quantitative study, the data could be organized in tables, I believe that it would be somewhat easier to make a comparison between the experiments.
Discussion: presents the comparison between the results of the three experiments; makes a discussion with references and also brings the conclusions of the study.
References are relevant and up-to-date.
Yours sincerely,
Author Response
Dear reviewer and Editor,
Thank you for your positive comments on our manuscript. According to your suggestions, we have organized the data in tables at the end of each experiment. The revisions have been marked up using the “Track Changes” function, you could view the detailed change in the new version of our manuscript. If there are any other modifications we could make, we would modify them and we really appreciate your help.
Yours sincerely

Reviewer 2 Report
This paper is well-written and deals with an important topic. Authors used three experiments to show the different moral judgments of artificial intelligence in different conditions.
The significant level should be labeled in the Figures.
The paper is very interesting to me as a scientist working in the artificial intelligence-related field.
This paper has numerous implications not only for artificial intelligence studies but also for the moral issues with human subjects.
Author Response
Dear reviewer and Editor,
Thank you for your positive comments on our manuscript. According to your suggestions, we have labeled the significant level in the Figures. The revisions have been marked up using the “Track Changes” function, you could view the detailed change in the new version of our manuscript. If there are any other modifications we could make, we would modify them and we really appreciate your help.
Yours sincerely

Reviewer 3 Report
Introduction
1.3. Artificial Intelligence and moral agents
Here, the authors talk about the need to study the human morality in relation to artificial intelligence.
In this sense, many recent studies have focused on the problem of endowing autonomous agents with morality, especially with reference to self-driving cars.
I suggest the authors to improve the state of the art on this topic, as AVs are a typical example of autonomous agents which is given a particular responsibility in the choice, improving this aspect could increase the necessity of understanding how much humans consider moral the actions of humans or artificial agents and this work could be better appreciated in view of its practical implications in today's technological development.
Experiment 2:
Here, and in the following experiment 3, there are the results in the introduction related to the previous experiment (in the experiment 2 authors talk about results of experiment 1; and in the experiment 3 authors talk about the results of experiment 2). This is not very clear to the reader.
I suggest writing the result at the end of any session (for example: Experiment 1: Participant; Materials; Design; Results) or just in the discussion.
Experiment 3:
It is not very clear if with “Within-subjects” authors intend that the participants are the same that did the test in the Experiment 1 and 2. Please clarify.
Discussion:
Here, authors hypothesize about the differences between experiment 1 (trolley dilemma) and experiment 2 (footbridge dilemma). The results of experiment 3 are not very clear. Specifically, authors say “in Experiment 1, the actual action did not influence people’s moral judgments in the trolley dilemma, while in Experiment 3, people rated the action as less moral and more morally wrong than inaction in the trolley dilemma”.
This sentence is not very clear. I understand that while in the first experiment (trolley dilemma) people are interested more in the difference between human and AI, in the third experiment related to the scenario that concern the trolley dilemma, people are more interested in the action vs inaction.
If that's correct, why does this happen? Could it be an effect of repeated testing on the same people? Or if people are different between experiments, could it be due to personal and cultural differences? What do the authors think?
Author Response
Dear reviewer and Editor,
We feel great thanks for your professional review work on our manuscript. As you are concerned, there are several problems that need to be addressed or improved. According to your nice suggestions, we have made extensive corrections to our previous draft, the detailed corrections are listed below. The reviewer’s comments are laid out below in italicized font. Our response is given in normal font.
The reviewer’s comment 1:
Introduction
1.3. Artificial Intelligence and moral agents
Here, the authors talk about the need to study the human morality in relation to artificial intelligence.
In this sense, many recent studies have focused on the problem of endowing autonomous agents with morality, especially with reference to self-driving cars.
I suggest the authors to improve the state of the art on this topic, as AVs are a typical example of autonomous agents which is given a particular responsibility in the choice, improving this aspect could increase the necessity of understanding how much humans consider moral the actions of humans or artificial agents and this work could be better appreciated in view of its practical implications in today's technological development.
The authors’ response 1:
Thanks for your kind suggestions, we have added a paragraph in section “1.3. Artificial Intelligence and moral agents”. According to your suggestions, we take AVs as an example to describe the practical implications of the current study. The revisions have been marked up using the “Track Changes” function, you could view the detailed change in the new version of our manuscript.
The reviewer’s comment 2:
Experiment 2:
Here, and in the following experiment 3, there are the results in the introduction related to the previous experiment (in the experiment 2 authors talk about results of experiment 1; and in the experiment 3 authors talk about the results of experiment 2). This is not very clear to the reader.
I suggest writing the result at the end of any session (for example: Experiment 1: Participant; Materials; Design; Results) or just in the discussion.
The authors’ response 2:
In our previous manuscript, we wrote the session “Materials and Methods” of Experiment 1~3 first and then wrote the results of Experiment 1~3 according to the template suggested by the journal, but we causes a problem of ambiguity and neglect the reader's reading experience. We sincerely thank you for your careful reading and your valuable feedback. We have revised our manuscript by writing the result at the end of any session according to your suggestion, you could view the detailed change in the new version of our manuscript.
The reviewer’s comment 3:
Experiment 3:
It is not very clear if with “Within-subjects” authors intend that the participants are the same that did the test in the Experiment 1 and 2. Please clarify.
The authors’ response 3:
Dear reviewer, we did not use a within-subjects design for the purpose of expecting the results of experiment 3 to be similar to those of Experiments 1 and 2. We used a within-subjects design in Experiment 3 for two reasons: 1) though we conducted Experiment 1 and 2 separately, and the participants in Experiment 1 and Experiment 2 were not the same people, but the procedures and measures are the same, so we think the Experiment 1 and 2 may have a similar effect to a between-subjects design. Therefore, we use a within-subjects design in Experiment 3; 2) the participants in Experiment 1 and Experiment 2 were not the same group, we cannot exclude the influence of individual differences, therefore, we conduct a within-subjects in Experiment 3 to do a repeated testing on the same people.
The reviewer’s comment 4:
Discussion:
Here, authors hypothesize about the differences between experiment 1 (trolley dilemma) and experiment 2 (footbridge dilemma). The results of experiment 3 are not very clear. Specifically, authors say “in Experiment 1, the actual action did not influence people’s moral judgments in the trolley dilemma, while in Experiment 3, people rated the action as less moral and more morally wrong than inaction in the trolley dilemma”.
This sentence is not very clear. I understand that while in the first experiment (trolley dilemma) people are interested more in the difference between human and AI, in the third experiment related to the scenario that concern the trolley dilemma, people are more interested in the action vs inaction.
If that's correct, why does this happen? Could it be an effect of repeated testing on the same people? Or if people are different between experiments, could it be due to personal and cultural differences? What do the authors think?
The authors’ response 4:
Dear reviewer, this sentence is the correct result, just as you understand, in Experiment 1(trolley dilemma) people are interested more in the difference between humans and AI, in Experiment 2 (footbridge dilemma) people are interested in the action vs inaction. In the third experiment related to the scenario that concerns the footbridge dilemma, the result was exactly similar to Experiment 2 (footbridge dilemma) that people are interested in the action vs inaction but not interested in the difference between humans and AI. But in the third experiment related to the scenario that concern the trolley dilemma, people are interested in both the difference between human and AI and the action vs inaction, it’s different from Experiment 1(trolley dilemma) where people are only interested in the difference between human and AI. So we use this sentence to point out the difference between Experiment 3 and Experiment 1, but we are sorry we did not write it clearly enough for the readers to understand, we have modified our expression in the latest version of our manuscript.
As for the reasons for the difference, we think it could be an effect of repeated testing on the same people in Experiment 3. In Experiment 1, participants only read the scenario that concerns the trolley dilemma, this scenario did not arise participants’ strong negative emotions, thus people are interested more in the difference between humans and AI but ignore the action vs inaction. While in Experiment 3, it’s a within-subjects design, participants read both the trolley dilemma and footbridge dilemma, we had half the subjects read the trolley dilemma first and another half the subjects read the footbridge dilemma, we thought this would eliminate the sequential effect, however, in fact, it’s possible that the participants were already influenced by the first situation when reading the second one. Therefore, we suspect that in Experiment 3, if participants read the footbridge dilemma first and are interested in the action vs inaction due to strong negative emotions, this may influence their subsequent concerns, they may also be interested in the action vs inaction when reading the scenario that concerns the trolley dilemma. Thank you for your careful reading and feedback, we are sorry we did not express this sentence clearly enough and did not explain why this happen. We have already revised the sentence and added this limitation of our experiments in the first and final paragraph of the discussion session in the latest version of our manuscript.
Thank you again for your comments and valuable suggestions to improve the quality of our manuscript, if there are any other modifications we could make, we would modify them and we really appreciate your help.

Reviewer 4 Report
The Authors investigate how people make moral judgments about the behavior of artificial intelligence agents in two main dilemma: the trolley and the footbridge one. In the first case, people are generally driven by controlled cognitive processes while, in the second option, people are generally driven by automatic emotional responses.
As a main outcome, the Authors found that, in the two different moral dilemmas, people adapt different modes of moral judgment to artificial intelligence. The reason for this relies on the fact that, when people make moral judgments in different types of moral dilemmas, they are engaging in different processing systems.
The manuscript contains original results and it is written in a very clear way. The research design is appropriate and the methodology employed is adequately described. Limitations and future directions are fairly highlighted in the Discussion session.
It meets the criteria of scientific quality and filed relevance for this journal. It is also suitably formatted for publication.
I recommend the manuscript for publication in the present form.
Author Response
Dear reviewer and Editor,
We appreciate it so much for your careful reading and your nice comments on our manuscript. Wish you all the best.
Yours sincerely
